# Possible Extracellular Signals to Ameliorate Sarcopenia in Response to Medium-Chain Triglycerides (8:0 and 10:0) in Frail Older Adults

**DOI:** 10.3390/nu16162606

**Published:** 2024-08-08

**Authors:** Osamu Ezaki

**Affiliations:** Institute of Women’s Health Science, Showa Women’s University, Tokyo 154-8533, Japan; ezaki1952@yahoo.co.jp

**Keywords:** acyl-ghrelin, beta2-adrenergic receptor, PGC-1alpha isoform, sympathetic nervous system, sarcopenia, muscle mass, medium chain fatty acid, medium chain triglyceride, ketone body, protein-sparing effect

## Abstract

In frail older adults (mean age 85 years old), a 3-month supplementation with a low dose (6 g/day) of medium-chain triglycerides (MCTs; C8:0 and C10:0) given at a meal increased muscle mass and function, relative to supplementation with long-chain triglycerides (LCTs), but it decreased fat mass. The reduction in fat mass was partly due to increased postprandial energy expenditure by stimulation of the sympathetic nervous system (SNS). However, the extracellular signals to ameliorate sarcopenia are unclear. The following three potential extracellular signals to increase muscle mass and function after MCT supplementation are discussed: (1) Activating SNS—the hypothesis for this is based on evidence that a beta2-adrenergic receptor agonist acutely (1–24 h) markedly upregulates isoforms of peroxisomal proliferator-activated receptor gamma coactivator-1alpha (PGC-1alpha) mRNAs, promotes mitochondrial biogenesis, and chronically (~1 month) induces muscle hypertrophy. (2) An increased concentration of plasma acyl-ghrelin stimulates growth hormone secretion. (3) A nitrogen-sparing effect of ketone bodies, which fuel skeletal muscle, may promote muscle protein synthesis and prevent muscle protein breakdown. This review will help guide clinical trials of using MCTs to treat primary (age-related) sarcopenia.

## 1. Introduction

Medium-chain triglycerides (MCTs; C8:0 and C10:0) are promising nutrients for the treatment of sarcopenia [1]. Recently, a combined data analysis of two clinical trials with frail older adults (mean age 85 years old) showed that relative to supplementation with 6 g long-chain triglycerides (LCTs)/day, supplementation with 6 g MCTs/day for 3 months statistically significantly increased body weight, body mass index (BMI), muscle mass (right arm muscle area, left calf circumference), and strength/function (right-hand grip strength, right and left knee extension times, walking speed, and number of iterations in leg open and close test) and decreased fat mass (right triceps skinfold thickness) [2]. These increases in favored measures of muscle mass and function were observed with and without supplementation with l-leucine (1.2 g/day) and vitamin D-enriched essential amino acids [2]. The significant increase in body weight in the MCTs-supplemented group is somewhat surprising because dietary MCTs have been considered to reduce fat mass and body weight (while not affecting muscle mass) relative to dietary LCTs [3,4,5,6,7]. The reduction in fat mass was partly due to increased postprandial energy expenditure by stimulation of the sympathetic nervous system (SNS) (Section 4.1). However, the mechanisms of increased muscle mass and function after MCT supplementation in frail older adults are unclear.

The increases in muscle mass and function after MCT supplementation might be due to the following characteristics of participants and protocols in the clinical trials with frail older adults [1,2,8] when compared with other MCT trials with healthy young adults [3,4,5,6,7]: (1) Participants in a nursing home were very old adults (mean age was 85 years old). The response to MCTs may differ in young and older muscle cells. (2) The chronic effects of MCT supplementation for 3 months with a meal based on dietary reference intake in Japan were examined and, therefore, they were unlikely to suffer from severe malnutrition. (3) An MCT dose of 6 g/day is much less than that which aims to increase ketone bodies in the blood (usually 20~40 g/day of MCTs) [9]. A 6 g/day intake of MCTs (75% C8:0 and 25% C10:0) corresponds to about 4% of habitual energy intake (overall mean habitual energy intake was 1430 kcal/day), 15% of habitual fat intake (overall mean fat intake was 39 g/day), and 0.14 g/kg body weight (overall mean body weight was 42.6 kg) in this population.

Marked progress has been made in understanding the etiology of sarcopenia in the past two decades, which may lead to new treatments for this condition. Peroxisomal proliferator-activated receptor gamma coactivator-1alpha (PGC-1alpha) is an inducible transcriptional coactivator of mitochondrial biogenesis and cellular energy metabolism in skeletal muscle that may ameliorate muscle atrophy [10]. A reduction in both RNA and protein expression of PGC-1alpha and downstream signaling targets in slow- and fast-twitch muscle fibers (i.e., multinuclear muscle cells containing fibers of actin and myosin) have been considered major causes of mitochondrial dysfunction during aging [11]. This mitochondrial dysfunction leads to the depletion of adenosine triphosphate (ATP) in muscle fibers, which may result in muscle atrophy [11,12].

The expression of the PGC-1alpha gene is regulated by two different promoters—proximal (canonical) and alternative [13,14,15]. A beta2-adrenergic receptor (AR) agonist activates the alternative promoter, not the proximal promoter [13]. Via a beta2-AR/cAMP/PKA/cAMP-response element-binding protein (CREB) signaling cascade [16] (Figure 1), an activation of beta2-AR led to increases in the isoforms of PGC-1alpha mRNAs (named PGC-1alpha-b and PGC-1alpha-c) in mouse [17] and human [18] skeletal muscles and PGC-1alpha protein in mouse skeletal muscle [15], and to an increase in mitochondrial function in C2C12 cells [19]. Chronically, beta2-AR agonist (formoterol) administration for 4 weeks in old rats attenuated age-related muscle wasting and weakness [20]. Therefore, MCT supplementation may ameliorate muscle atrophy by way of SNS activation.

This review outlines dietary MCTs’ fate (Section 2) and summarizes previous reports describing the mechanisms of MCTs’ effects on the amelioration of muscle atrophy (Section 3). Then, taking the characteristics of the clinical trials’ participants and protocols into consideration, the following three possible extracellular signals causing ameliorating effects of MCTs on primary sarcopenia are discussed: (1) SNS activation (Section 4), (2) acyl-ghrelin (Section 5), and (3) the nitrogen-sparing effect of ketone bodies (Section 6).

## 2. Outline of Metabolisms in Dietary MCTs

Dietary MCTs can hydrolyze to medium-chain fatty acids (MCFAs) by lingual lipase in the oral cavity and stomach [21]. MCFAs in the stomach are used to form acyl-ghrelin, which can bind and activate the growth hormone secretagogues receptor (GHSR) [22]. Discovered in the stomach in 1999, ghrelin requires C8:0 to bind and activate GHSR [23]. Ghrelin O-acyltransferase (GOAT) acylates proghrelin with C8:0 to produce acyl-proghrelin, which is converted to acyl-ghrelin by the prohormone convertase [22,24]. Other MCFAs (C6:0 or C10:0) may substitute for C8:0 [25]. MCFAs in the intestinal lumen reduce the concentration of plasma gastric inhibitory peptide (GIP), which leads to decreased fat mass [26,27]. Note that increased concentrations of GIP promote obesity [28]. MCFAs absorbed in the intestine are transferred to the liver via the portal vein and metabolized to acetyl-CoA in the mitochondria [29]. Surplus production of acetyl-CoA that neither enters the TCA cycle nor participates in fatty acid synthesis leads to the formation of ketone bodies (beta-hydroxybutyrate [BHB] and acetoacetate). MCFAs are metabolized more rapidly and completely than LCFAs [30]. This rapid oxidation leads to greater ketone body formation. MCFAs that escape metabolism in the liver and ketone bodies produced by the liver (ketone bodies are not metabolized in the liver) are distributed in the blood via the circulatory system [31]. Peripheral tissues such as the brain, skeletal muscle, and heart can use both MCFAs [32,33] and ketone bodies [34,35] as energy sources and signaling molecules.

## 3. Previous Studies Elucidated Cellular Mechanisms for Amelioration of Muscle Atrophy in Response to MCT Supplementation

The mechanisms of MCTs’ ameliorating effects on muscle atrophy have been elucidated using animal models of energy restriction [36], immobilization [37], and cachexia [38,39] with their appropriate controls.

Three-week-old male Wistar rats were malnourished under a 40% energy-restricted diet containing either MCTs (65% C8:0 and 19% C10:0; MCTs corresponding to 16% of total energy intake and 86% of total fat intake) or LCTs for 15 days and were then sacrificed [36]. The total skeletal muscle weight of the right leg tended to be 15% higher in the MCT-fed group than in the LCT-fed group (*p* = 0.10) [36]. Plasma albumin and insulin concentrations were higher in the MCT-fed mice than in the LCT-fed mice. A two-fold increase in plasma insulin concentration in the MCT-fed group may have contributed to increased protein synthesis in the liver and muscles.

Nine-week-old male Wistar rats fed a diet containing either MCTs (60% C8:0 and 22% C10:0; MCTs corresponding to 14% of total energy intake and 86% of total fat intake) or LCTs for 3 days underwent unilateral hindlimb immobilization [37]. Rats were sacrificed after immobilization for 3, 7, and 14 days. The immobilized soleus muscle mass in the LCT-fed rats decreased markedly compared to that in the contralateral muscle; however, these losses were partially suppressed in the MCT-fed mice with an inhibition of increased muscle ring-finger protein (MuRF-1) content in immobilized muscle. These data suggest that dietary MCT partially alleviated immobilization-induced muscle atrophy by inhibiting the ubiquitin-proteasome pathway.

Seven-week-old male ICR mice were fed a regular chow diet containing either MCT oil (MCFAs not specified) or LCT oil for 1 week [38]. Then, lipopolysaccharide (LPS) was injected into the mice, and 24 h later, they were killed and plasma and tissues were obtained. MCT supplementation suppressed the LPS-induced decrease in plasma ketone bodies and citrate synthase activity (a marker of mitochondrial enzyme activity) in tibialis anterior muscles but did not affect the LPS-induced reduction in PGC-1alpha protein in tibialis anterior muscles relative to LCTs. The authors suggested that the amount of PGC-1alpha protein might not change for 1 week of supplementation.

Six-week-old male Sprague–Dawley rats received a gastric tube, and 48 h later, LPS was injected [39]. Then, rats received enteral nutrition (EN, 100 kcal/kg body weight) containing different doses of C8:0 (0, 0.25, 0.5, and 1 g/kg body weight, respectively) via the gastric tube for 3 days. After subcutaneous injection of L-^13^C-valine to measure the fractional synthetic rate of muscle proteins, rats were sacrificed. LPS led to decreased body weight and insulin resistance and increased serum inflammatory cytokines, inhibited AKT, inhibited mechanistic target of rapamycin complex (mTORC)1, inhibited forkhead box protein O (FOXO)-1 phosphorylation in muscle (extensor digitorum longus), increased the expression of atrogin-1 and muscle ring-finger protein (MuRF)-1 in muscle, and decreased the fractional synthetic rate of muscle proteins. C8:0-rich EN and simple EN (a dose of 0 g/kg body weight of C8:0) significantly ameliorated these LPS-induced cachexia phenotypes. Importantly, C8:0-rich EN showed a pronounced ameliorating effect compared with simple EN and also increased acyl-ghrelin in circulation, both in a dose-dependent manner.

In a cell experiment in C2C12 myotubes, MCFA (C10:0 or C12:0)-treated cells displayed decreased lipid accumulation, increased mitochondrial oxidative capacity, and increased amounts of PGC-1alpha protein compared with LCFA (C16:0, C18:1, or C18:2)-treated cells [40].

These results indicated that MCT supplementation under several pathophysiological conditions could increase muscle mass and function relative to appropriate controls, with promoted mitochondrial biogenesis, increased protein synthesis, and inhibition of protein degradation. However, aged model mice and their upstream signals (extracellular signals) have yet to be examined.

## 4. Sympathetic Nervous System Activation

Acute SNS activation promotes mitochondrial biogenesis via increased SNS-sensitive PGC-1alpha isoform [13,41]. Note that muscle mass did not increase in transgenic mice with skeletal muscle-specific overexpression of either PGC-1alpha-a (canonical form), PGC-1alpha-b (isoform), or PGC-1alpha-c (isoform), although they showed mitochondrial biogenesis [41]. Instead, 25-week-old PGC-1alpha overexpressing transgenic mice showed muscle atrophy with depletion of ATP by way of the uncoupling of oxidative phosphorylation [42]. Promoting mitochondrial biogenesis may prevent or treat muscle atrophy but does not increase muscle mass under normal conditions.

### 4.1. MCT Supplementation Increases Energy Expenditure during the Postprandial Period by Activating the SNS

In healthy mostly overweight adults, MCTs combined with energy-restricted diets resulted in greater fat mass loss and body weight loss than did LCTs combined with energy-restricted diets, and their mechanisms have been elucidated [43,44,45,46,47]. However, the energy restriction per se markedly alters the metabolism in the whole body (e.g., decreased plasma insulin and glucose concentrations, increased plasma ketone bodies and free fatty acids concentrations, and others), which may affect the impact of MCTs on energy metabolism [7]. Therefore, clinical trials to examine the effect of MCT supplementation with a regular diet (but not with an energy-restricted diet) on energy expenditure were informative. Repeating a single meal (1000 kcal) containing MCTs (60% C8:0 and 30% C10:0, 400 kcal in each meal) on a regular diet for 5 days showed a two-fold postprandial energy expenditure than that containing LCTs, suggesting that under conditions favorable for fat synthesis (i.e., higher insulin concentrations), MCTs have a greater thermic effect of food than LCTs [48]. In addition, a greater thermic effect of food following a single meal (400–900 kcal) containing MCTs was also observed compared with a meal containing LCTs [49,50]. Interestingly, a respiratory chamber study showed that the intake of MCTs (15–30 g per day) as part of a habitual diet enhanced daily energy expenditure by about 500 KJ/day, increasing 24-h levels of urinary noradrenaline but not adrenaline and dopamine [51].

In the animal studies that examined the effects of C8:0 and C10:0, an increase in norepinephrine concentration was observed in mice fed an MCFA-enriched diet (MCFAs corresponding to 4% of total energy intake and 10% of total fat intake, a dose similar to that of clinical trials with frail older adults) compared with mice fed an LCFA-enriched diet, which leads to increased lipolysis in white and brown adipose tissues and thermogenesis in brown adipose tissue with reductions in body weight and fat mass [52,53].

These results strongly suggest that, compared to LCT supplementation, MCT supplementation taken with a meal increases energy expenditure during the postprandial period by activating the SNS.

### 4.2. Activation of SNS May Ameliorate Muscle Atrophy in Primary Sarcopenia

Increasing the PGC-1alpha protein in skeletal muscle is one promising strategy for ameliorating primary sarcopenia. Although exercise is recommended for this purpose, it is challenging for frail older adults. However, SNS activation can increase the PGC-1alpha protein without exercise [15].

As described in Section 1, a beta2-AR activation markedly increases the expression of isoforms of PGC-1 alpha mRNA in skeletal muscle via the alternative promoter [13,17,54]. SNS-induced isoforms of PGC-1alpha (i.e., PGC-1alpha-b and PGC-1alpha-c) mRNA are expressed in skeletal muscles, and they are functional [41,55,56]. Of 795 amino acids in the total canonical PGC-1alpha protein, only the N-terminal 16 amino acids differed from those in PGC-1alpha-b or PGC-1alpha-c [13]. Recently, to examine the effects of the PGC-1alpha derived from the alternative promotor on whole-body metabolism, mice disrupting the alternative exon 1 were produced [57]. As expected, these mice failed to upregulate muscle PGC-1alpha mRNAs in response to exercise and cold exposure, and eventually developed obesity and hyperinsulinemia. These data suggest that the PGC-1alpha-b and PGC-1alpha-c isoforms respond to external stimuli and are crucial for energy metabolism in the whole body.

In animal and human studies, chronic administrations of beta2-AR agonists, which are prescribed as bronchodilators for asthmatic patients, increased skeletal muscle mass and decreased fat mass [16,58]. Their mechanisms were elucidated as follows: activating the AKT/mTORC1 signaling pathway [59,60], attenuating the myostatin signaling pathway [61], and reducing calpain activity [60]. The muscle hypertrophy requires a stimulatory G-protein subunit [62] and beta-arrestin 1 [63]. However, in some studies, chronic treatment of a beta2-adrenergic receptor agonist (clenbuterol) resulted in decreases in mitochondrial contents and fatty acid oxidation [64,65], which were associated with a reduction of PGC-1alpha mRNA and protein [64]. Therefore, it is conceivable that beta2-AR agonists could increase muscle mass independent of the PGC-1alpha protein [16]. The acute effects of beta2-agonists may fundamentally differ from their chronic effects. The mechanisms by which the acute effects of a beta2-AR agonist are transient are unclear but may be partly due to a reduction in beta2-AR concentration in skeletal muscle by a long-time beta2-AR agonist administration (so-called “desensitization of beta2-AR signaling”) [66].

### 4.3. Conclusions

MCT supplementation activates the SNS, which may prevent the decreased muscle mass and function observed in primary sarcopenia. A signal transduction pathway of beta2-AR to increase transcription of the PGC-1alpha isoform is shown in Figure 1. Note that the subcutaneous injection of clenbuterol in rats phosphorylates and activates AKT in skeletal muscles, by which muscle protein synthesis is promoted via PKA-independent signaling pathways [59]. Charlot et al. reported that, under normal conditions, 12-week-old male C57BL/6J mice fed a C8:0-rich diet (C8:0 corresponding to 18% of total energy intake and 80% of total fat intake) for 6 weeks presented higher spontaneous activity and endurance capacities, along with promoted mitochondrial biogenesis in skeletal muscle with increased PGC-1alpha mRNA and protein levels, relative to mice fed a chow (LCFAs-rich) diet [67]. However, SNS activity and body composition in each group of mice were not measured.

To prove the contributions of the SNS to the effect of MCTs on the prevention of sarcopenia, as well as animal studies examining the mechanisms using beta2-AR knockout mice, several clinical studies of MCT supplements targeted at older adults are needed to examine the chronic effects of MCTs on the association between postprandial SNS activity and muscle mass. If the results were positive, the tissues or soluble factors responsible for SNS activation might be sought. As suggested by cell culture experiments [40], MCFA might directly activate muscle fibers via beta2-adrenergic receptors. Recently, it was reported that the activation of the neurons in the ventromedial hypothalamic nucleus expressing steroidogenic factor-1 increases the sympathoadrenal activity and isoforms of PGC-1alpha in skeletal muscles via multiple downstream nodes in mice [68]. Therefore, MCT supplementation may activate the central nervous system to activate the SNS, increasing muscle mass and function. In addition, clinical trials to determine a suitable cessation interval for MCT supplementations might be necessary to avoid the possible desensitization of beta2-AR signaling due to the continuous activation of the SNS.

## 5. Acyl-Ghrelin

Although many fatty acids exist in nature, MCFAs (C6:0, C8:0, and C10:0) are specific; they are part of acyl-ghrelin (active ghrelin or simply ghrelin), an orexigenic hormone produced by the stomach [23]. Without MCFAs, no acyl-ghrelin is produced. Acyl-ghrelin stimulates GH release, and GH increases muscle mass (the MCTs/Ghrelin/GH hypothesis).

### 5.1. Clinical Trials to Estimate Muscle Mass in Response to MCTs, Acyl-Ghrelin, and GHSR Agonists

In clinical trials examining the effects of MCTs, acyl-ghrelin, and GHSR agonists on muscle mass, GHSR agonists consistently showed increased muscle mass [69,70,71]. However, clinical trials of MCTs were insufficient as the intervention periods were very short, and no negative control (i.e., LCTs) was provided.

Acyl-ghrelin (mostly C8:0-ghrelin) produced in the stomach enters the blood circulation or activates the gastric afferent vagal nerve and stimulates GH release in the pituitary [72]. GH can transmit its anabolic signal directly through its receptor or indirectly through insulin-like growth factor 1 (IGF-1) stimulation. Primary GH and IGF-1 receptor downstream signaling, which increases muscle mass, occurs via AKT [73] (Figure 1). To investigate this hypothesis, blood acyl-ghrelin concentrations after MCT supplementation were measured, and all studies showed increases in acyl-ghrelin concentration from its baseline value [74,75,76,77]. In one of these studies, body weight was significantly increased for a 2-week treatment: a single oral ingestion of 3 g of MCT (100% C8:0) with an EN formula (400 kcal in 400 mL) after breakfast in cachectic patients increased plasma acyl-ghrelin, and the 2-week administration of MCTs with the formula increased appetite score, body weight, and serum albumin and IGF-1 concentrations [74]. However, these increases after MCT supplementation were relative to their baseline values but not those in the LCTs-supplemented formula group. The increased body weight may be due to increased energy intake from the EN formula. The other three studies also did not make comparisons with an LCT group (isocaloric negative control) [75,76,77].

It is well known that acute acyl-ghrelin injection increases GH concentration, appetite, and food intake and that chronic acyl-ghrelin injection increases body weight [72]. However, few studies have shown increases in muscle mass, possibly due to the shorter intervention periods. Chronic effects of acyl-ghrelin over years are mimicked by GHSR agonists, which have been reported in three clinical trials [69,70,71], all of which showed increases in body weight and fat-free mass with no change in fat mass. Note that GH has a strong lipolytic effect on adipose tissues [78]; therefore, the GHSR agonist should have decreased fat mass. The reason for this discrepancy is unclear at present.

### 5.2. Animal Study to Support the MCTs/Ghrelin/GH Hypothesis

In an additional experiment by Zhang et al., to further examine the effects of acyl-ghrelin on LPS-induced cachexia (see Section 3), a specific inhibitor of GOAT (Go-CoA-Tat) was added to C8:0-rich EN [39]. The protective effects of C8:0-rich EN against cachexia were abolished mostly by the administration of Go-CoA-Tat, along with a decrease in serum acyl-ghrelin concentration. These results suggested that increased serum acyl-ghrelin causes the protective effects of dietary C8:0 against cachexia.

### 5.3. Conclusions

Applying the results from LPS-injected animals [39], “increased acyl-ghrelin concentration” appeared to be a cause of MCTs’ effects in ameliorating primary sarcopenia. However, the effects of MCTs in animal models of primary sarcopenia, using inhibitors of GOAT or mice with knockout of GOAT, GHSR, or ghrelin are needed. Postprandial plasma acyl-ghrelin concentrations after MCT supplementation should be measured in frail older adults, who eventually develop increased muscle mass and function in 3-month intervention periods.

## 6. Nitrogen-Sparing Effect of Ketone Bodies

In a clinical trial of frail older adults (mean age 86.6 years) in a nursing home, LCT (6 g/day) supplementation with l-leucine (1.2 g/day), l-isoleucine (0.3 g/day), l-valine (0.3 g/day), and other essential amino acids and vitamins including a vitamin D (20 μg/day)-enriched supplement combined with a regular diet for 3 months did not significantly increase muscle mass and function relative to no supplementation [8]. The branched-chain amino acids (leucine, isoleucine, and valine) in circulation were preferentially used by muscle for energy production [79]. Leucine has been well known to play a pivotal role in the protein-sparing effects of amino acids [80]. Also, leucine located in the cytosol in muscle fiber indirectly activates the mTORC1, which is a protein kinase that phosphorylates substrates to potentiate anabolic processes and inhibits catabolic ones [81].

This result suggested that the participants in this nursing home might not be severely malnourished. Larger amounts of leucine and other essential amino acids or longer feeding periods (more than 3 months) may be required to obtain significant increases in muscle mass and function. In other words, even under this condition (i.e., non-significant effects of 1.2 g/day of leucine supplementation), 6 g MCTs/day of supplementation with a meal can effectively increase muscle mass and function. Because MCTs induce ketogenesis (Section 2), ketone bodies might be the molecules responsible for increases in muscle mass and function.

### 6.1. Clinical Trials Supporting the Nitrogen-Sparing Effect of MCT Relative to LCT Supplementation

Krotkiewski reported that in obese women (n = 22 in each group, overall mean age 43 years) receiving a very low-calorie diet (579 kcal/day), the MCT group (9.9 g MCTs/day) increased the rate of decline of body fat (MCT −5.8 kg vs. LCT −4.9 kg after 4 weeks, *p* < 0.05) but decreased the rate of decline of fat-free mass (MCT −2.7 kg vs. LCT −3.2 kg after 4 weeks, *p* < 0.05) relative to the LCT groups [43]. The author suggested that the increased plasma ketone body concentrations caused by dietary MCTs may provide additional energy to muscles and prevent the breakdown of muscle tissues caused by malnutrition (nitrogen-sparing effects of ketone bodies) [43]. In another study of clinically ill patients with sepsis and trauma receiving total parenteral nutrition (TPN), an improving effect of a structured MCT-containing emulsion (n = 9, mean age 52 years, 1.5 g MCTs/kg body weight/day) on the nitrogen balance was observed for at least 3 days of administration compared to an LCT emulsion (n = 11, mean age 59 years, 1.5 g LCTs/kg body weight/day). Thus, it appears that the nitrogen-sparing effect of ketone bodies was observed with long-term low-calorie diets or protein-energy malnutrition. The frail older adults who increased muscle mass and function after MCT supplementation might be experiencing some malnutrition; therefore, the nitrogen-sparing effect of ketone bodies might be in operation.

### 6.2. Can Increased Ketone Bodies after MCT Supplementation Promote Muscle Protein Synthesis?

Postprandial plasma ketone body concentrations after MCT supplementation with a meal have not been measured in clinical trials in frail older adults [1,8] but have been estimated by similar studies. For example, in healthy young volunteers, after oral intake of 10 g MCT (60% C8:0 and 40% C10:0) in a complete liquid diet mimicking a standard meal, high plasma levels of ketone bodies were produced over 2 h with a maximum plasma total ketone bodies concentration of 409 μM (150 μM at baseline) [82]. In another study, a supplementation with 10 g MCTs (60% C8:0 and 40% C10:0)/day at breakfast increased blood ketone body concentrations from 49–97 μM (baseline) to 455 μM (maximum) 30 min after MCT supplementation, followed by a return to baseline levels after 3 h [9]. Therefore, we estimated that 6 g MCT supplementation with a meal might result in mild ketosis (~0.5 mM) during a 3-h postprandial period.

The nitrogen-sparing effect of ketone bodies in skeletal muscle may depend on the acute (with or without a meal) and chronic (lean or obese) energy status in the whole body. Several clinical studies examined the acute effect of exogenous ketone bodies on muscle protein synthesis under different feeding conditions. Without a meal, which favors ketogenesis, more than 1 mM concentrations of ketone bodies seemed necessary to observe a meaningful nitrogen-sparing effect [83,84]. A 3-h infusion of BHB revealed nitrogen conservation in the whole body without a meal when total blood ketone body concentrations were 1.1–1.2 mM [83]. In another study, after an overnight fast, an 8-h infusion of BHB without a meal increased the incorporation of a tracer C^13^ leucine into skeletal muscles (protein synthesis) by 5–17% and decreased the appearance of the C^13^ in the expired CO_2_ (leucine oxidation) by 18–41% compared to that during normal saline infusion, during which plasma BHB concentrations were 1–2.5 mM [84].

Very recently, the effects of acute ingestion of exogenous ketone monoester with and without dietary protein (10 g whey protein) co-ingestion on postprandial myofibrillar protein synthesis rates have been examined in healthy young males [85]. In a 1-h postprandial period, blood BHB concentrations reached 3.2–3.4 mM in the ketone-only and the ketone plus protein groups, whereas the concentration in the protein-only group was 0.6 mM. In all three groups, the postprandial rate of muscle protein synthesis was higher than that in basal conditions, with no difference between the groups. This result indicated that a large amount of plasma ketone bodies did not promote an increase in the rate of muscle protein synthesis either in the absence or presence of a protein-rich meal.

Taken together, 6 g MCTs/day supplementation with a regular meal, which might induce mild ketosis (~0.5 mM), may not exhibit the nitrogen-sparing effect of ketone bodies. However, the chronic effects of ketone bodies have yet to be examined. Repeated small increases in ketone bodies at each meal may contribute to the chronic increase in muscle mass in frail older adults.

### 6.3. Possible Molecular Mechanisms of the Protein-Sparing Effects of Ketone Bodies

The molecular mechanisms are speculated as follows. Under malnutrition, due to a shortage of ATP and increased adenosine monophosphate (AMP) in skeletal muscle, AMP-activated protein kinase (AMPK), the sensor of intracellular energy levels and glucose availability, is activated [86]. Then, AMPK reduces mTORC1 activity and inhibits protein synthesis [87]. Increases in ketone bodies could reverse these steps by providing ATP to skeletal muscles (Figure 1). However, it is worth noting that if a large amount of ketone bodies and fatty acids were given, AMPK in skeletal muscle may be activated due to a shortage of glucose.

Sirtuin 1 (SIRT1), an enzyme of class III deacetylase, may relate to ketone bodies’ protein-sparing effects. In cells, it is well known that SIRT1 is activated by an increased nicotinamide adenine dinucleotide (NAD)^+^/NADH ratio, which plays an important role in cell survival and longevity under caloric restriction [88]. That may also apply to humans. When blood glucose declines due to a low-carbohydrate diet, the inhibited glycolysis increases the amount of NAD^+^ and then activates SIRT1. In skeletal muscle, to utilize fatty acids instead of glucose as a fuel, SIRT1 deacetylates and activates the PGC-1alpha protein and promotes mitochondrial biogenesis [89]. SIRT1 also increases mitochondrial biogenesis by increasing the expression of mitochondrial transcription factor A (TFAM) in a PGC-1alpha-independent manner [90]. In addition, SIRT1 deacetylates and inactivates FOXO and inhibits muscle atrophy [91]. Therefore, SIRT1 activation might increase muscle mass and function [92].

Ketone bodies may activate SIRT1 in some tissues. The use of ^31^P phosphorus magnetic resonance spectroscopy (MRS) in humans showed that a 10 g MCT (60% C8:0 and 40% C10:0) supplementation increased the amount of NAD^+^ in the brain concomitant with an increase in plasma BHB concentration [82]. Increased NAD^+^ may activate SIRT1. Cockayne syndrome is a progressive neurodegenerative disorder with mitochondrial dysfunction caused by mutations in genes encoding DNA repair proteins. In a mouse model of this condition, a high-fat diet increased blood and brain BHB concentrations, activated brain SIRT1, and rescued their phenotypes, including hearing loss but not muscle function [93]. In addition, in this cell model, BHB treatment activated SIRT1 and rescued its associated phenotypes [93]. Therefore, BHB can activate SIRT1 in the brain and change the phenotypes. However, whether BHB activates SIRT1 in skeletal muscle and whether it increases muscle mass and function are still being determined.

### 6.4. Conclusions

To prove the nitrogen-sparing effect of ketone bodies in frail older adults, the chronic effects of supplementation with a low dose of ketone monoester on muscle protein synthesis rate and muscle mass should be investigated in this population. Positive results indicate the need to clarify the molecular mechanisms of the nitrogen-sparing effects of ketone bodies.

## 7. Limitation

The mechanisms of MCTs’ ameliorating effects on primary sarcopenia described here were mostly based on experiments using young animals and tissues. It is conceivable that young cells and older cells respond to MCTs differently. Therefore, comparative studies of MCTs targeted at young and older cells, tissues, animals, and humans are recommended.

## 8. Conclusions

Based on recent progress in understanding the etiology of primary sarcopenia and the discovery of SNS-sensitive isoforms of PGC-1alpha, three possible extracellular signals causing increases in muscle mass and function after MCT supplementation in frail older adults were discussed, as follows: (1) activation of the SNS, (2) an increased plasma acyl-ghrelin concentration, and (3) a nitrogen-sparing effect of ketone bodies. Each hypothesis is possible but will require further research that includes clinical trials. Studies using appropriate knockout mice and specific inhibitors will also be necessary to clarify the causality.

## Figures and Tables

**Figure 1 nutrients-16-02606-f001:**
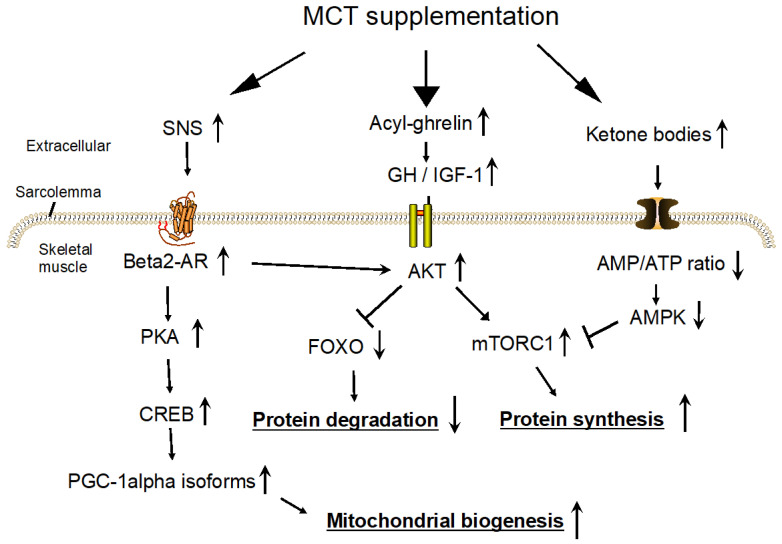
Three possible signaling pathways causing increased muscle mass and function in response to supplementation with medium-chain triglycerides (C8:0 and C10:0) in frail older adults. They are, from left to right: (1) SNS activation, (2) acyl-ghrelin, and (3) a nitrogen-sparing effect of ketone bodies. Note that each of their main signaling pathways is shown here. See the text for detail. The lines with arrows indicate activation, whereas lines with flat ends indicate inhibition between two molecules. The upward line after the word represents the activated state and the downward line indicates the inhibited state after MCT supplementation relative to controls. AMP, adenosine monophosphate; AMPK, AMP-activated protein kinase; AR, adrenergic receptor; CREB, cAMP-response element-binding protein; FOXO, forkhead box protein O; GH, growth hormone; IGF, insulin-like growth factor; MCT, medium-chain triglyceride; mTORC, mechanistic (mammalian) target of rapamycin complex; PKA, protein kinase A; PGC, peroxisomal proliferator-activated receptor gamma coactivator; SNS, sympathetic nervous system.

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
