# Peer review of "Possible Extracellular Signals to Ameliorate Sarcopenia in Response to Medium-Chain Triglycerides (8:0 and 10:0) in Frail Older Adults"

_nutrients, 2024, doi:10.3390/nu16162606_

Round 1
Reviewer 1 Report
Comments and Suggestions for Authors
This manuscript entitled Possible Extracellular Signals to Ameliorate Sarcopenia in Response to Medium-Chain Triglycerides (8:0 and 10:0) in Frail Older Adults. It was well-written by author. I have several comments for this manuscript:
1) The study should address the limitations and offer some future research direction for researchers.
2) Manuscript language should be improved.
Line 33, "muscle mass (right arm muscle area, left calf circumference),..." why did the author only consider the right arm and left calf measurements? Please clarify it.
Many included studies were enrolled older adults aged 85+ years, but the age in rats was 3-9 weeks. Despite the fact that different experimental subjects may have different cell growth stages, such as young and old cells, the authors should explain these physiological changes in terms of how they affect muscle function. It may consider as limitation.
Line 324, "In a clinical trial of frail older adults in a nursing home,..." Please add the age of older adults.
Line 397-398, "Repeated small increases in ketone bodies at each meal may contribute to the chronic increase in muscle mass in frail older adults." Please add relevant references to support your point.
Manuscript language should be improved.
Author Response
Reviewer 1: This manuscript entitled Possible Extracellular Signals to Ameliorate Sarcopenia in Response to Medium-Chain Triglycerides (8:0 and 10:0) in Frail Older Adults. It was well-written by author. I have several comments for this manuscript:
Response: Thank you very much for reviewing this manuscript.
- The study should address the limitations and offer some future research direction for researchers.
Response: Yes, the limitations and some future research directions are described in a new Section 7 as follows.
" 7. Limitation The mechanisms of MCTs' ameliorating effects on primary sarcopenia were mostly based on experiments using young animals and tissues. It is conceivable that young cells and older cells respond to MCTs differently. Therefore, comparative studies of MCTs targeted at young and older cells, tissues, animals, and humans are recommended." (lines 442-446)
Also, in Section 1, the sentence "The response to MCTs may differ in young and older muscle cells." was added in line 49.
- Manuscript language should be improved.
Response: Yes, the corrected text is shown in red font. The acknowledgment was deleted because it was obvious and not necessary.
- Line 33, "muscle mass (right arm muscle area, left calf circumference),..." why did the author only consider the right arm and left calf measurements? Please clarify it.
Response: There is a misunderstanding. We considered the right and left arm and the right and left calf.
We measured the right and left arm muscle area and calf circumferences at baseline and after intervention in the MCTs and the LCTs groups. In each measurement, after the intervention, the change from baseline was larger in the MCTs-containing groups than in the LCTs-containing groups. However, a statistically significant difference in changes from baseline value between the two groups was observed in measurements of the right arm muscle area and left calf circumference but not in those of the left arm muscle area and the right calf circumference (Table 3 in ref. 2.). For the arm muscle area, dominant hands were right in most of the participants. This may be a reason. However, the reason for the non-statistical significance of the change in right calf circumference between the groups is unclear. This might be due to the small number of participants. Because some of these have been discussed in the original report (ref 1) and were not related to this review, I did not discuss them in this review.
- Many included studies were enrolled older adults aged 85+ years, but the age in rats was 3-9 weeks. Despite the fact that different experimental subjects may have different cell growth stages, such as young and old cells, the authors should explain these physiological changes in terms of how they affect muscle function. It may consider as limitation.
Response: The physiological and molecular changes in muscle cells during aging might be an important topic in cell biology, and their interaction with MCT supplementation might be discussed. However, this discussion is beyond the current manuscript. I would like to discuss these in the future. The limitation is shown in Section 7 (lines 442-446).
- Line 324, "In a clinical trial of frail older adults in a nursing home,..." Please add the age of older adults.
Response: Yes, age of older adults was added as follows "(mean age 86.6 years)" (line 325).
- Line 397-398, "Repeated small increases in ketone bodies at each meal may contribute to the chronic increase in muscle mass in frail older adults." Please add relevant references to support your point.
Response: Unfortunately, I cannot find relevant references to support this hypothesis.
Reviewer 2 Report
Comments and Suggestions for Authors
This review deals with one of the major concerns of aging: is it possible to prevent, or at least restrain, sarcopenia with intervention on diet? The Author analyzes literature about the response of medium-chain tryglicerides in olders in cell signaling.
The manuscript is exploring deeply this issue, it is worthable to be publish as a review only after all the sentences in which the Author starts with I are changed in impersonal way: i.e. -line 77- I described must be changed in this review were described... and so on
Comments on the Quality of English LanguageThis manuscript is a really good review but needs minor editing
Author Response
Reviewer 2: This review deals with one of the major concerns of aging: is it possible to prevent, or at least restrain, sarcopenia with intervention on diet? The Author analyzes literature about the response of medium-chain triglycerides in olders in cell signaling.
- The manuscript is exploring deeply this issue, it is worthable to be publish as a review only after all the sentences in which the Author starts with I are changed in impersonal way: i.e. -line 77- I described must be changed in this review were described... and so on.
Response: Thank you very much for your favorable response. The sentences in the text were changed to be impersonal.
- This manuscript is a really good review but needs minor editing.
Response: Thank you again. Some errors were corrected and shown in red font in the text. The acknowledgment was deleted because it was obvious and not necessary.